# Brain Vascular Health in ALS Is Mediated through Motor Cortex Microvascular Integrity

**DOI:** 10.3390/cells12060957

**Published:** 2023-03-21

**Authors:** Stefanie Schreiber, Jose Bernal, Philipp Arndt, Frank Schreiber, Patrick Müller, Lorena Morton, Rüdiger Christian Braun-Dullaeus, Maria Del Carmen Valdés-Hernández, Roberto Duarte, Joanna Marguerite Wardlaw, Sven Günther Meuth, Grazia Mietzner, Stefan Vielhaber, Ildiko Rita Dunay, Alexander Dityatev, Solveig Jandke, Hendrik Mattern

**Affiliations:** 1Department of Neurology, Otto von Guericke University Magdeburg, Medical Faculty, 39120 Magdeburg, Germany; 2German Center for Neurodegenerative Diseases (DZNE) within the Helmholtz Association, 39120 Magdeburg, Germany; 3Center for Behavioral Brain Sciences (CBBS), 39106 Magdeburg, Germany; 4Department of Internal Medicine/Cardiology and Angiology, Otto von Guericke University Magdeburg, 39120 Magdeburg, Germany; 5Institute of Inflammation and Neurodegeneration, Otto von Guericke University Magdeburg, 39120 Magdeburg, Germany; 6Centre for Clinical Brain Sciences, The University of Edinburgh, UK Dementia Research Institute Centre, Edinburgh EH16 4UX, UK; 7Department of Neurology, Medical Faculty, Heinrich Heine University Düsseldorf, 40225 Düsseldorf, Germany; 8Medical Faculty, Otto von Guericke University, 39120 Magdeburg, Germany; 9Department of Biomedical Magnetic Resonance, Faculty of Natural Sciences, Otto von Guericke University Magdeburg, 39120 Magdeburg, Germany

**Keywords:** amyotrophic lateral sclerosis, vascular supply, motor cortex, pericytes, exerkines

## Abstract

Brain vascular health appears to be critical for preventing the development of amyotrophic lateral sclerosis (ALS) and slowing its progression. ALS patients often demonstrate cardiovascular risk factors and commonly suffer from cerebrovascular disease, with evidence of pathological alterations in their small cerebral blood vessels. Impaired vascular brain health has detrimental effects on motor neurons: vascular endothelial growth factor levels are lowered in ALS, which can compromise endothelial cell formation and the integrity of the blood–brain barrier. Increased turnover of neurovascular unit cells precedes their senescence, which, together with pericyte alterations, further fosters the failure of toxic metabolite removal. We here provide a comprehensive overview of the pathogenesis of impaired brain vascular health in ALS and how novel magnetic resonance imaging techniques can aid its detection. In particular, we discuss vascular patterns of blood supply to the motor cortex with the number of branches from the anterior and middle cerebral arteries acting as a novel marker of resistance and resilience against downstream effects of vascular risk and events in ALS. We outline how certain interventions adapted to patient needs and capabilities have the potential to mechanistically target the brain microvasculature towards favorable motor cortex blood supply patterns. Through this strategy, we aim to guide novel approaches to ALS management and a better understanding of ALS pathophysiology.

## 1. Introduction

Amyotrophic lateral sclerosis (ALS) is a progressive disease involving motor neurons. A neuropathological hallmark of ALS is associated with cytoplasmic inclusions in degenerating motor neurons consisting of abnormally ubiquitinated and phosphorylated transactive response DNA-binding protein of 43 kDa (pTDP-43), which can be found in the motor cortex particularly early in the course of disease [1]. Motor neurons have a high energy consumption rate due to significant intracellular metabolic demand. They are therefore vulnerable to oxidative stress, mitochondrial dysfunction, hyperexcitability, and glutamate-mediated excitotoxicity, which are considered upstream events promoting pTDP-43 aggregation, and thus, the initiation and clinical manifestation of ALS (please see [2] for a comprehensive review article on this subject). Neuronal pTDP-43 aggregation occurs with loss of physiological cellular functions, such as DNA damage repair and axonal transport mechanisms, which leads to neurodegeneration.

Clinically, degeneration of cortical motor neurons first affects one distinct area within the motor homunculus unilaterally, which is reflected by the focality of initial motor phenotypes, classified as bulbar, left or right upper or lower limb onset [3]. The degeneration of cortical motor neurons then spreads sequentially from lateral to medial within the motor homunculus of the ipsilateral hemisphere (bulbar, upper limb, lower limb) or from the ipsi- to the contralateral hemisphere (same limb). Accordingly, clinical motor symptoms rather evolve from rostral to caudal regions [3,4]. Afterwards, pTDP-43 pathology extends to adjacent neocortices (prefrontal, postcentral and temporal), the basal ganglia and the medial temporal lobe (MTL) [5].

Although ALS is a rare disease, the number of affected patients worldwide is expected to increase on average by 69% (range 33–116%) over the next 20 years [6]. Only around 5–10% suffer from hereditary (familial) ALS, while the remaining 90–95% of disease cases have a sporadic, late(er) onset form. The underlying risk or resistance factors for the sporadic forms remain poorly understood [7]. The same is true for disease progression, with only a few clinical variables considered as predictors of a more rapid decline (e.g., weight loss, bulbar onset, diagnostic delay and cognitive impairment) [8,9]. The disease is highly heterogeneous and the duration of survival varies markedly among individuals, although the majority of ALS patients die within 3–5 years of symptom onset. Indeed, some patients survive up to 10 years after diagnosis, and some appear to have halted or even reversed disease progression [10,11]. This indicates that there must be factors and mechanisms that are heterogeneous between patients.

We here outline the concept of brain vascular health on the molecular, cellular and organ level in ALS and discuss how brain vascular health could be mediated through vascular patterns of blood supply to the motor cortex and serve as a potential marker of resistance and resilience against downstream effects of vascular risk and events in ALS. We present novel magnetic resonance imaging (MRI) techniques to detect brain vascular health and propose new targeted therapies for its maintenance and recovery. Finally, we advertise strategies for the practical management of ALS patients to implement vascular rethinking in the clinical setting.

## 2. Vascular Health for Reserve

The evolving and dynamic field of brain reserve, maintenance and compensation in healthy cognitive aging has gained a lot of interest due to its role in understanding inter-individual variability in cognitive performance. The possibility of enhancement (neural capacity and efficacy), preservation (neuronal repair and plasticity) and recruitment of neuronal resources to improve cognition is of clear societal relevance and interest [12]. Here, we specifically refer to the concept of “resistance”, i.e., the avoidance of pathology, and “resilience”, i.e., tolerance against the effect of pathology (“coping”) [13]. Thus far, research in these areas has been applied primarily to healthy aging and to (preclinical) Alzheimer’s disease (AD), where education, lifestyle or behavioral modifications (e.g., intellectual engagement throughout the lifespan, physical activity) explain a large proportion of the variance in cognition [13,14]. In ALS, there is an urgent need to clarify similar factors and mechanisms of resistance and resilience, which might allow for new preventive and therapeutic strategies for this incurable neurodegenerative disorder, which is currently limited to palliative treatment [15,16].

Just recently, it has become clear that preservation of systemic vascular health in aging, i.e., low cardiovascular risk and minimal cardiovascular disease leads to sustained normal cognition over time by maintaining brain metabolism [17]. Likewise, a genetic risk for vascular disease is related to lower resilience of cognitive performance in aged individuals with different degrees of AD pathology [18]. Several comprehensive reviews and position papers have emphasized the interaction between cognitive function in aging, brain injury and systemic vascular health. In this context, risk factors such as arterial hypertension and obesity appear to have a significant impact on cognitive function in later life if they are present in midlife and persist [19,20,21,22]. Systemic vascular health thus exerts its effects on the brain over decades rather than years, allowing a significant time period for prevention and treatment. Additionally, brain vascular cell abnormalities are increasingly recognized as early and important contributors to the pathophysiology of several neurodegenerative diseases, including AD, frontotemporal dementia, early-onset dementia and Huntington’s disease [23,24,25,26]. The relationship between systemic vascular health and brain health is likely mediated by mechanisms of resistance and resilience. Brain vascular health can therefore be recognized as a target that could be tackled over a long period of time to achieve greater resistance and resilience against neurodegenerative disease [27].

We propose that systemic and brain vascular health could be added to the emerging concept of the exposome in ALS, i.e., the role of cumulative environmental lifetime exposures interacting with the patient’s genetic risk (gene–time–environment hypothesis) as they seem to be central to ALS risk and disease progression [28,29,30].

The relationship between vascular health and ALS is supported by several clinical observations and systematic studies. First, ALS patients suffer from cerebrovascular disease with significantly higher frequency than control subjects [31]. Depending on the geographic location, up to half of ALS patients have arterial hypertension and up to a quarter coronary artery disease [32]. Additionally, polygenetic ALS risk correlates with smoking status, physical inactivity, obesity and a poor blood lipid profile, favoring systemic vascular disease [33,34,35]. Secondly, ALS patients with comorbid cardiovascular risk factors and disease show more rapid decline and potentially shorter survival than ALS patients without [36]. Conversely, treatment of vascular risk factors, particularly arterial hypertension and diabetes, reduces the risk of ALS disease onset [37]. Furthermore, ALS patients display small vessel abnormalities affecting multiple organ systems, including the brain, retina, muscle and skin [38,39,40,41].

In addition, the structure and function of the motor cortex and white matter seem to additionally be highly vulnerable to poor vascular health. A greater change in blood pressure in aging, for example, has been associated with lower corticospinal tract (CST) integrity, which in turn predicts worse downstream motor control [42]. Further, small infarcts and microinfarcts, which are ischemic lesions related to cerebrovascular disease, are commonly localized in the motor cortex and the vicinity of the CST [43,44,45]. In aged individuals with an increased vascular risk profile, CST dysfunction further relates to perivascular spaces (PVS) in the juxtacortical white matter of the motor cortex, which are known markers of microvascular disease in the brain [43].

## 3. Vascular Supply Mediates Brain Vascular Health

More than two thirds of the general population display variations of the circle of Willis (CoW) that deviate from the expected anatomy [46]. CoW variations lead to different types of vascular supply to the brain and have been determined for the anterior and posterior circulation as well as for hippocampal and lenticulostriate arteries [47,48,49].

### 3.1. Vascular Supply of the Medial Temporal Lobe

Just recently, we investigated the circulation of the aging MTL, which can be distinguished by the number of large arteries contributing to the vascular supply. The small terminal feeding vessels originated either from only one, i.e., “single supply” or from two, i.e., “double supply”, large arteries (posterior cerebral artery and/or—if existent—uncal branches from the anterior choroid artery). Participants with a double-supplied MTL displayed better preserved temporal lobe and whole-brain gray matter volume, as well as better sustained memory and global cognitive function [50,51,52]. These results suggest that patterns of vascular supply can be considered a measure of resistance and resilience that can be observed in brain regions beyond the supplied core area. The morphology of the CoW, i.e., its collateral supply, is also associated with the occurrence and type of vascular brain disease, such as hemorrhagic or ischemic lesions, which further supports the idea that vascular patterns of supply represent a potential marker of resistance [53,54]. Evaluation of these parameters could identify individuals at low/high risk and be of value to the development of new strategies against the initiation and manifestation of MTL-dominant neurodegeneration, such as in aging or AD.

### 3.2. Vascular Supply of the Motor Cortex

Variable vascular supply has also been observed in the human motor cortex. The medial motor cortex is supplied by downstream vessels of the anterior cerebral artery (ACA, pericallosal and callosomarginal artery), while the lateral motor cortex is supplied by vessels originating from the medial cerebral artery (MCA, precentral, central and postcentral artery group) [55]. One postmortem study determined the vascular supply pattern in each hemisphere separately for the medial and the lateral motor cortex in 20 adults. A single supply pattern (i.e., supply by one ACA or one MCA downstream branch) was more prevalent than a double supply pattern (i.e., supply by more than one ACA or more than one MCA downstream branch), for both the medial (72.5% vs. 27.5%) and the lateral motor cortex (82.5% vs. 17.5%) [55].

Leveraging the high-resolution capabilities of 7 Tesla (T) MRI, preliminary data demonstrated that it is possible to reproduce the postmortem results (see Figure 1). To that end, MPRAGE data was acquired at 7T (Siemens Healthineers, Erlangen, Germany) with 0.45 mm isotropic resolution for 19 subjects (without any known neurological pathology; seven females; 31.18 ± 6.48 years old; given written, informed consent). Besides enabling identification of the motor cortex, MPRAGE data acquired at 7T provides hyperintense arteries. Thus, with the high resolution used, the vessels of interest were delineated to determine the vessel pattern per hemisphere. In accordance with the postmortem study, a single supply pattern was more prevalent in the medial motor cortex (68.4% vs. 31.6%; 34.2% pericallosal artery dominance; 34.2% callosomarginal artery dominance), while the preliminary MRI data showed a balanced prevalence of single and double supply (50.0% vs. 50.0%; 34.2% central group dominance; 15.8% precentral group dominance) in the lateral motor cortex.

This observed variability in distinct vascular supply patterns of the ACA and MCA could drive the subject-specific overlap of vascular territories within as well as between the lateral and medial motor cortex. Hence, as for the MTL, one could anticipate that these vascular supply patterns might be associated with motor function and could provide insight into mechanisms of resistance and resilience interacting to ALS development. This assumption is indirectly supported by few studies, revealing (i) different effect sizes for the relationship between cortex perfusion (assessed through single photon emission tomography (SPECT)) and bulbar, upper or lower limb function in ALS, and (ii) motor cortical hypoperfusion (assessed through arterial spin labeling (ASL)) in ALS (which correlated with overt motor function) without motor cortical atrophy.

Mechanistically, we speculate that, in the case of double supply, an overlapping perfusion territory of the feeding arteries might preserve adequate cerebral blood flow (CBF), small vessel density and microvascular function, ensuring optimal neuronal nutrition and removal of toxic metabolites from the brain. In addition, brain areas with a single vascular supply might have less capacity to compensate for the effects of long-term exposure to vascular risk factors, thus leading to early impaired microvascular function, small vessel wall remodeling, accelerated microvascular aging, and development of lesions. Lesions, e.g., microinfarcts, are commonly found in all motor cortex layers and have been considered to follow the pattern of layer-specific arteriolar blood supply, which further suggests that vessel morphology and distribution are markers of resistance to disease [45,56]. Experimentally, cortical microinfarcts exerted long-distance effects on the fiber integrity of white matter tracts, linking vascular supply to (widespread) neurodegeneration [57].

### 3.3. Vascular Supply—Molecular and Cellular Underpinnings in ALS

Accordingly, in ALS there are several lines of evidence that molecular and cellular alterations of the central nervous system (CNS) microvasculature play a pivotal role at preclinical and clinical disease stages (Figure 2).

ALS patients frequently harbor mutations in the vascular endothelial growth factor (VEGF) gene; (motor cortical) protein expression and VEGF serum levels are reduced in ALS rodent models and patients [58,59,60]. VEGF exerts trophic and neuroprotective effects on motor neurons [61]. One hypothesis to explain these protective effects is that VEGF promotes a “vascular niche”. After secretion from motor neurons and endothelial cells, VEGF fosters endothelial cell formation, pericyte proliferation and migration, and thus, small vessel wall integrity. This guarantees locally optimized oxygen and energy supply and removal of toxic metabolites, protecting motor neurons from oxidative stress and glutamate-mediated excitotoxicity, and thus, from pTDP-43-mediated neurodegeneration (please see [62] for review). Accordingly, experimental VEGF reduction has been related to accelerated motor neuron degeneration and disease onset, whereas experimental VEGF overexpression has been shown to have the opposite effect, i.e., prolonged motor neuron survival and delayed disease onset [59,63,64]. The pertinence of this experimental intervention is supported by the fact that a phase II clinical trial has been approved and initiated, applying intracerebroventricular recombinant VEGF in ALS patients [65] (ClinicalTrials.gov NCT01384162 (accessed on 7 February 2023)).

Furthermore, capillary pericyte coverage and integrity are reduced in ALS models and patients, albeit some (co-)effects of aging or secondary disease have to be considered as well [41,66,67]. Pericytes promote blood–CNS barrier integrity, regulate CBF and cerebrovascular reactivity (CVR), and are involved in toxic metabolite removal [68]. In ALS, pericyte deficiency and functional loss goes along with a reduction of small vessel density, collapsed small vessel lumen (through wall contraction) and aberrant angiogenesis [69,70,71,72]. Subsequent CBF/CVR reduction fosters insufficient oxygen and energy supply and could hinder toxic metabolite removal, which precedes and promotes motor neuron degeneration [72]. Hence, in experimental ALS, intraperitoneal injection of pericytes increased survival, while co-culture between pericytes and motor neurons/other neuronal cells elevated the expression of genes encoding antioxidant enzymes, which both could point towards slowing of motor neuron degeneration [73].

### 3.4. Factors Downstream of Cell Activation in the Neurovascular Unit

Experimental studies in transgenic ALS mice have uncovered that microvascular cells, such as endothelial cells, pericytes, and perivascular fibroblasts, are already activated in asymptomatic and preclinical disease stages. This activation occurs with increased cellular metabolic demand, turnover and wall repair, likely indicating a period of compensation in early disease stages, which indicates the significant contribution of small vessel wall integrity to initial disease dynamics [74]. Most likely, chronic cellular activation favors the premature senescence of vascular cells, i.e., cellular inability to proliferate, regenerate, and resist stress and apoptosis, finally resulting in loss of neurovascular unit (NVU) cells. This concept has already emerged in the aging process of cerebral vascular cells and especially in AD [23,75]. Senescent cells develop a senescence-associated secretory phenotype (SASP) characterized by a set of proinflammatory cytokines, chemokines, reactive oxygen species, growth factors and proteases, and interact through certain receptors (e.g., intercellular adhesion molecule-1 (ICAM-1)) and insoluble factors (e.g., fibronectin, collagen). SASP leads to immune cell recruitment, particularly via interleukin (IL)-6 and IL-8 secretion, and overall creates a local microenvironment that promotes senescence dissemination, i.e., imparts stress on surrounding cells, and chronic (inflammatory) tissue damage, which in turn drives chronic low-grade neuroinflammation and finally, neurodegeneration [76,77]. Indeed, immunosenescence, e.g., through reduction of mitochondrial energy production in chronically activated astrocytes and microglia has been established in ALS pathophysiology [78,79].

In experimental ALS and in autopsy tissue of patients, activated perivascular fibroblasts have been found in PVS, where they secrete SPP1 (osteopontin). In ALS patients, higher SPP1 serum levels predicted shorter survival, linking microvascular alterations to prognosis [74,80]. Early vascular cell perturbations and impairment of microvascular function might also lead to clearance failure of neurotoxic metabolites. This has been shown in an ALS rodent model and could be a further contributing factor to abnormal pTDP-43 accumulation, which is supported by the notion that, in ALS, pTDP-43 also deposits in cerebral small blood vessel walls [81]. Conversely, elevated levels of pTDP-43 in motor neurons, on the other hand, affect microvascular integrity via activation of NVU cells and leakage of the blood–brain barrier (BBB), thereby creating a self-reinforcing cycle of vascular malfunction and neurodegeneration [82].

Activation of microglia following BBB leakage may lead to remodeling and even degradation of the extracellular matrix in the adjacent neuropil areas [83,84], which plays an important role to protect highly active neurons from the oxidative stress, particularly in the form of perineuronal nets associated with GABAergic interneurons and a subset of pyramidal cells [85,86]. This might be an important factor for progression of neurodegeneration. This hypothesis is supported by studies showing (i) dysregulation of spinal chondroitin sulfate proteoglycans forming perineuronal nets in an early symptomatic superoxide dismutase (SOD)1^G93A^ transgenic rat model of ALS, and (ii) elevated degradation of perineuronal nets in this model after treatment with protease ADAMTS4 (a secreted disintegrin and metalloproteinase with thrombospondin motifs 4 normally expressed by several cell types including microglia), resulting in increased motoneuronal degeneration [87,88].

Illustrated alterations of the NVU are well in line with the huge number of experimental and human postmortem studies reporting leakage of the blood–brain and blood–spinal cord barrier in ALS (please see [89,90] for a detailed review). Several of the experimental and autopsy data are derived mainly from the spinal cord and the brainstem in ALS, thus demonstrating molecular and cellular microvascular alterations only in areas of the nervous system containing lower motor neurons. Damage of the NVU, however, takes place in a similar way in the human motor cortex, as already shown in an autopsy ALS study conducted in 1984 [91]. Strikingly, there is a lack of in vivo imaging studies focusing on vascular alterations in the human ALS motor cortex. This is in marked contrast to the substantial number of human structural imaging studies on motor cortex and CST degeneration in these patients.

## 4. In Vivo Imaging of Brain Vascular Health in the Motor Cortex

Further to anatomical, functional, and diffusion MRI to study ALS in vivo and non-invasively [92,93], MRI provides the potential to characterize microvascular alterations and vessel supply patterns in a temporospatial manner, over the course of disease progression. However, this potential is yet to be fully realized [94]. Currently, especially in the clinical setting, brain microvascular health is largely defined as the absence of structural MRI lesions related to small vessel pathologies, such as white matter hyperintensities, lacunes and cerebral microbleeds. These lesions, however, only represent the longstanding downstream consequences of microvascular alterations, while it is widely recognized in the vascular field that subvisible changes are present well before lesions appear. This challenge is thus not unique to ALS research, but a general challenge of translating available cutting-edge methods to routine application. Further, there is a gap between studies working on understanding disease pathogenesis at the microscopic/cellular level and routine clinical MRI studies providing macroscopic information regarding brain structure. From an empirical point of view, the BBB (here defined through leakage of contrast medium) in ALS patients seems to be intact on routine clinical MRI, compared, e.g., to the BBB in CNS tumors or primary neuroinflammatory disease, where BBB disruption is typically demonstrated by gadolinium extravasation [95]. However, studies providing data at the microscopic scale indicate microvascular alterations and BBB breakdown in ALS [89]. Hence, when focusing on preclinical and early stages of ALS, MRI-based markers reflecting more subtle changes in the microvascular health more directly are required. Ultra-high resolution at the mesoscopic scale (100–500 µm) is achievable using MRI and provides a potential bridge between the micro- and macroscopic scale. Further to a link between cellular processes and downstream consequences, such as macroscopic lesions detected with ultra-high-resolution MRI, vascular supply patterns can be assessed in vivo to understand potential mechanisms of resistance and resilience in ALS.

### 4.1. MR Markers of Microvascular Health

Compared to structural lesions, emerging measures, such as CVR [94,96,97,98,99], BBB integrity, perfusion and clearance [100,101,102,103,104,105,106,107,108], are more closely tied to the vasculature and could bridge the gap from (early) microvascular to brain injury [27]. Note that these techniques do not measure the response of individual microvessels but estimate the net response per volume element at the macroscopic scale. Each of these markers can be assessed with different MRI methods. For BBB integrity, perfusion and clearance, approaches can be broadly stratified by their use of exogenous, i.e., intravenous or intrathecal injection of gadolinium compounds [95,109,110,111,112], or endogenous tracer, i.e., magnetically-labeled blood water [100,101,107]. For CVR, approaches can be stratified by paradigm and MR sequence used. To achieve a change in arterial CO2 partial pressure, which is required to measure CVR [97], there are several paradigms. Most commonly, set-ups with external devices to control breathing gas composition, breath-hold tasks, and free breathing without any instructions are used [94,96,97,98]. MR data acquisition is commonly performed using either Blood-Oxygenation-Level Dependent (BOLD) functional MRI (fMRI) or ASL, although other MR sequences are also applicable [97]. For all these vascular markers, the most sensible implementation, i.e., MR acquisition and processing strategy, depends on the study at hand, i.e., subject compliance, installed MR sequences and available hardware. A detailed comparison of the available implementation for each marker can be found in the above-referenced literature. However, the main challenge to facilitate these novel markers is their translation into routine clinical practice.

In the context of microvascular function, clearance and BBB breakdown, PVS are a topic of current investigation. These fluid-filled spaces form between the two basement membranes of small arteries and are generally viewed to reflect impaired vascular function. PVS are found in several CNS pathologies [113], are readily available from routine clinical T1-weighted or, preferentially, T2-weighted images and are a complementary information to the abovementioned markers of microvascular health as well as downstream damage in brain structure. However, in ALS, PVS are currently solely described in the postmortem spinal cord, while for all other pathologies, frequent PVS in cerebral regions, such as the basal ganglia and centrum semiovale, are reported [74,113]. Our own preliminary data derived from 3T MRI of a large monocentric ALS cohort (diagnosis based on the revised El-Escorial criteria [114]; *n* = 161; 61 females; 61.9 ± 11.5 years old; given written informed consent; recruited through the neuromuscular outpatient clinic of the Department of Neurology, Otto von Guericke University in Magdeburg, Germany) show, for the first time, that these patients indeed display a significantly greater PVS burden in vivo compared to age- and sex-matched healthy controls (*n* = 49; 18 females; 61.3 ± 10.3 years old) as well (see Figure 3).

### 4.2. Ultra-High-Resolution MRI to Assess Vascular Supply Patterns

Moving from measuring the net effect of microvascular alterations at the millimeter level towards imaging of individual feeding vessels at the submillimeter level, recent advancements have enabled visualization of pial arteries as well as deep gray matter perforators [118,119,120,121] with spatial resolution of up to 0.14 mm isotropic. To that end, ultra-high field (UHF) MRI (B0 > 7T) was used as it provides increased signal-to-noise ratios compared to clinically available field strengths, enabling higher image resolution [122,123]. Additionally, with increasing main magnetic field strength, T1-relaxation times are prolonged [124]. This gives Time-of-Flight angiography an additional boost beyond the pure gain in signal-to-noise when using UHF. Therefore, with UHF MRI at the mesoscopic scale, it becomes possible to image feeding arteries completely non-invasively and in vivo (see Figure 1). This enables a novel, data-driven approach for identifying supply territories of arteries, as with decreasing vessel diameter the perfusion territory of an individual artery becomes more spatially specific [125,126]. When reaching the terminal branches, the perfusion territory per artery can be approximated as spherical [125]. Hence, the supply region of the arteries feeding the motor cortex can be approximated as the summation of all spherical-shaped territories of its higher order branches and therefore, the supply territory of a feeding artery is characterized by the distribution of and distance to its branches. This data-driven approximation of arterial supply territories from ultra-high-resolution MRI can be achieved by the recently proposed vessel distance mapping (VDM) technique [127,128,129]. Briefly, after segmenting the vessels from the MR images, VDM computes the Euclidian distance for each non-vessel voxel to its closest artery. By computing these distance maps for individual vessels, i.e., the ACA and MCA, and combining the vessel-specific maps by root-mean-square, a single map reflecting the closeness of each voxel to the feeding arteries is generated. While this is a pure data-driven approach and not a true perfusion measurement, VDM has proven insightful for understanding resilience of the hippocampus in cerebral small vessel disease [127,128,129]. In ALS, one could hypothesize that cortical regions with low distances to the feeding arteries are more resistant to ALS-induced microvascular and neurodegenerative pathology than regions with high distances. This is motivated by the fact that the proximity to multiple feeding arteries could be associated with a protective overlap of perfusion territories ensuring microvascular integrity and ongoing repair, and therefore optimal oxygen and energy supply and removal of toxic metabolites (see Figure 2).

### 4.3. MR-Based Assessment of Resistance and Resilience in ALS

ALS patients often present with complex and heterogenous clinical disease manifestations, with high variability in the affected muscle group at onset, the relative mix of upper motor neuron and lower motor neuron deficits and rate of progression. Clinically, all ALS patients show motor cortical motor neuron degeneration but can present different initial motor phenotypes classified as bulbar, left or right upper or lower limb onset [3].

For an individual patient, spreading patterns are variable and complex, which complicates counseling and prognosis. The described patterns of resistance and resilience by vascular mechanisms putatively determine the individual onset foci in the motor cortex and indicate regions of high vulnerability that are more likely to be affected early in the disease course (Figure 2B). In the MR-based assessment, these vulnerable motor cortical regions are marked by relatively high VDM levels and low vascular reserve. In contrast, areas of the motor cortex that have putatively protective double supply patterns and low levels of VDM would be more resistant and resilient to the spread of neurodegeneration. While the spread of ALS pathology appears to be diverse and complex, it might be the result of an ordered spatial progression shaped by the underlying individual vascular landscape that can be visualized using MRI.

Measuring the described vascular features may thus offer the unique possibility to more precisely predict the individual future spreading pattern and disease dynamic, leading to an improvement in clinical prediction and selection for clinical studies.

## 5. Targeting Brain Vascular Health to Preserve Microvascular Integrity in ALS—Emergent Concepts

Targeting the maintenance of vascular supply, i.e., assembling and restoring vessel patterns of favorable, e.g., double motor cortex supply, can be considered an innovative approach for establishing and fostering resistance and resilience in ALS. Indeed, one major advantage of acting on motor cortical vascular supply is the fact that these patterns can be longitudinally measured in human ALS and serve as a monitoring variable and proxy for brain microvascular health in clinical trials. In the following paragraph, we will discuss targeted (therapeutic) concepts to optimize perfusion and vascularization, which in turn could enhance microvascular collateralization, small vessel structure and function to move towards the development of advantageous supply patterns. Of note, the capability of microvascular repair is closely related to (motor)neuronal and synaptic plasticity and thus linked to resistance and resilience against neurodegeneration [130,131]. And, vice versa, preserved neuronal activity reinforces pericyte-mediated capillary dilation and thus, locally optimized CBF increase [69]. In the following sections, we will discuss maintenance and repair of the microvascular wall with regard to exerkines, pericyte restoration, senotherapeutics and cell-derived extracellular vesicle (EV)-based therapy as targeted approaches to enhance motor cortical vascular supply.

### 5.1. Physical Activity and Exercise

Physical activity is a low-cost intervention in primary and secondary prevention of numerous chronic diseases (e.g., dementia, diabetes and heart failure) [132]. In ALS, the beneficial role of physical activity and exercise remains controversial (reviewed in [133]), as there is limited evidence for prevention and treatment [134]. Several case-control studies with small numbers of ALS patients indicate a negative relationship between physical activity and the risk of disease development as well as disease progression [135]. In particular, professional soccer and American football players experience an increased incidence of ALS [136,137]. Potential explanations include head traumas and/or high intensity-associated oxidative stress levels. However, in a European case-control study with 652 ALS patients, no negative relationship was found [138]. Furthermore, animal research and exercise interventions with small sample size indicate that low-to-moderate physical exercise can improve the performance capability of ALS patients [139]. A current meta-analysis including seven randomized controlled trials with 322 patients showed that physical exercise can improve functional ability and pulmonary function of ALS patients [140].

Several epidemiological, observational and randomized controlled studies have shown positive effects of physical activity on cognition and reduced risk of neurodegenerative diseases [126,127]. However, the underlying neurobiological mechanisms of exercise-induced neuroprotection and neuroplasticity are still largely unknown. One potential mechanism might be based on the effects of peripherally secreted exerkines.

### 5.2. Exerkines

Exerkines are defined as signaling substances released into the blood from different tissues in response to exercise, e.g., from skeletal muscle (myokines) or the brain itself (neurokines). They exert beneficial effects on vascular brain health through endocrine (affecting distant tissue and fostering inter-organ communication) or autocrine and paracrine (affecting the cell of origin and adjacent ones) pathways [141]. Several myokines, such as VEGF, angiopoietin, nitric oxide or irisin have local (paracrine) effects on skeletal muscle vessel formation and perfusion and on endothelial function and microvascular tone. Via the endocrine effects of myokines, there is also a cross-talk between muscles and brain vascular health, in which the cerebral microvasculature and particularly, endothelial cells are discussed as central mediators of the beneficial effects [141]. Irisin, for example, crosses the BBB, and enriches in the brain after peripheral delivery. In various animal models, it reduces BBB leaks through inhibition of matrix metalloproteinases, peripheral immune cell infiltration and microglial activation [142,143,144]. However, in contrast to the growing experimental and clinical evidence that exercise has beneficial effects on synaptic plasticity, and neuronal and cognitive function in aging and neurodegenerative disorders, such as (preclinical) AD or Parkinson’s disease (PD), investigating its impact on microvascular integrity and motor function remains so far an understudied field [142,145,146,147].

A recent meta-analysis considering randomized controlled trials of aerobic and resistance exercise training in ALS showed that patients benefit in terms of motor function, increasing their resilience to disease [148]. However, as the majority of ALS patients suffer from progressive widespread motor symptoms, several forms of effective exercise such as aerobic training become increasingly inaccessible during the course of disease. ALS patients would therefore specifically benefit from the application of exerkines as therapeutics, i.e., “exercise in a pill”, to circumvent the difficulty of executing training programs [141]. Indeed, as stated above, there is an ongoing phase II clinical trial in ALS examining the effect of continuous intracerebroventricular VEGF delivery on motor function and survival using subcutaneous implanted pumps [65] (ClinicalTrials.gov NCT01384162, accessed on 7 March 2023). VEGF is an exerkine that directly stimulates endothelial cell proliferation, survival and microvascular wall stabilization. It also has neurotrophic effects, which was the main motivation to initiate its application within the clinical ALS trial [149]. Just recently, VEGF has been shown to additionally act on the tone of CNS lymphatic vessels, which promotes the clearance of senescent NVU cells from the brain [150]. Interestingly, in the VEGF phase II trial, ALS patients undergo repeated MRI including MR angiography and cerebrospinal fluid (CSF) sampling. Depending on the available field strength and sequences or storage procedure for CSF, these data could retrospectively be analyzed with respect to vascular brain health, comprising supply patterns and NVU integrity, measuring factors of resilience.

### 5.3. Pericyte Restoration

Increased development of pericyte-targeted therapies is predicted within the next 10 years. Targeting pericyte dysfunction has the potential to restore microvascular integrity, tone, perfusion and clearance [69,151]. Pharmaceutical interventions targeting brain pericytes have already been approved. Administration of calcium-channel blockers such as nimodipine can be considered a very practical approach, as the preferred anti-hypertensive treatment in ALS. It may reduce small vessel wall constriction, thereby increasing perfusion and preventing pericyte degeneration and BBB leakage, a mechanism that also has been shown in experimental stroke models [69]. Cilostazol, a phosphodiesterase type 3 inhibitor (antiplatelet) already in clinical use for obstructive large artery disease, has experimentally shown to reduce pericyte detachment from endothelial cells, promote pericyte proliferation and protect pericytes against apoptosis, thereby maintaining BBB integrity [152,153]. Intracerebroventricular application of recombinant human platelet-derived growth factor β (PDGF-β) has further entered one clinical trial in PD [154]. Physiologically, endothelial cell secretion of PDGF-β is essential for pericyte recruitment and proliferation, and thus for formation and maintenance of a functional BBB [155].

### 5.4. Senotherapeutics

Senotherapeutics reduce cellular senescence and may present an innovative strategy to treat incurable diseases. They are currently being tested in several clinical trials for neurodegenerative diseases, mainly AD [76]. Senotherapeutics either suppress the SASP (senomorphics) or promote death of senescent cells (senolytics). Senolytic targets include B-cell lymphoma 2 protein family members (Bcl-2), the phosphatidylinositol 3-kinase-related kinase family (PI3K), heat shock protein 90 (Hsp90), the cyclin-dependent kinase inhibitors p16 and p21, or the cell cycle inhibitor p53, which are all upregulated in astrocytes/glial cells in experimental and human ALS [76,156]. In humans, current senotherapeutics do not clear specific senescent CNS cell types such as those of the NVU, but instead act on senescent burden, mainly alleviating the overall proinflammatory microenvironment. Experimental modeling, however, confirmed that the senescent NVU is indeed targeted by senotherapeutics like dasatinib and quercetin (targeting, e.g., p16, p21, SASP), which are currently under investigation in several clinical trials focusing on AD, cancer or chronic kidney disease. Dasatinib and quercetin preserve the structure of the BBB, especially the expression of tight junction proteins [157]. Nevertheless, further upregulated targets in senescent NVU cells need to be identified before senotherapy can evolve from a promising general to a precise approach for rescuing microvascular dysfunction in ALS.

### 5.5. Cell-Based Therapies and Extracellular Vesicles

In experimental ALS models, transplantation of human bone marrow-derived (hBM) mesenchymal stroma cells and endothelial progenitor cells has been shown to promote repair and stabilization of the CNS microvascular wall, which includes an increase in pericyte coverage and leads to improved motor neuron survival and motor function (extensively reviewed in [89,158]). Experimental application of hBM mesenchymal stroma, endothelial or muscle progenitor cells further increased VEGF expression, suggesting to combine the advantages of cell-based and exerkine approaches (reviewed in [158]). Intriguingly, endothelial progenitor cells additionally release beneficial EVs, to strengthen and repair the microvascular wall in experimental ALS [159].

Despite increasingly convincing preclinical evidence for the potential of cell-based therapies for NVU function and microvascular repair, clinical trials using hBM-derived cells in ALS patients have focused on monitoring the (downstream) rescue of degenerating motor neurons (reviewed in [158]). Hence, the clinical role of hBM-induced (upstream) microcirculatory recovery as a potential mediator of motor neuron survival has not yet been elucidated. This might be partially explained by the current absence of sensitive tools for repetitive in vivo monitoring of CNS microvascular alterations in patients. From the perspective of motor cortex vascular supply, a recently initiated phase I clinical trial (CNS10-NPC-GDNF Delivered to the Motor Cortex for ALS—Full Text View—ClinicalTrials.gov, accessed on 7 March 2023) is of particular relevance. Here, cell-based therapy is used, in which neural progenitor cells producing VEGF are directly transplanted into the motor cortex of ALS patients [158,160]. Thus, therapy probably acts on the local microvasculature, which would provide the opportunity to monitor in situ the relationship between intervention, vascular supply, motor neuron degeneration and motor function with cutting-edge MRI sequences, as described above. Neural progenitors derived from induced pluripotent cells also protected perineuronal nets around the preserved motoneurons in SOD1^G93A^ transgenic rats [88]. Additionally, in a recent human trial in ALS, transient BBB opening was achieved using non-invasive MRI-guided focused ultrasound, which facilitates the transfer of therapeutics from the blood to the parenchyma and supports the feasibility of locally-targeted treatment and the subsequent possibility for in situ monitoring [161].

Overall, cell-based therapies are an encouraging approach. They have been translated from preclinical to clinical trials and offer a promising potential to target ALS pathogenesis, presumably including the involved NVU. Considering that a less invasive intravenous administration seems to be as clinically beneficial as intrathecal application, a general acceptance in the clinical setting should be achievable [152].

EVs are membrane-enclosed carriers of damage-associated molecular patterns secreted by all types of cells and are considered biofluid markers of, e.g., NVU cell activation and dysfunction. They are secreted as a tool of cellular communication by shuttling molecules that require protection from extracellular enzymatic degradation or that lack a signal sequence, such as microRNAs (miRNAs), lipids, cytokines and chemokines [162]. Based on their shuttling function and their ability to pass the BBB and avoid an immune response, EVs could be used as vehicles to treat microvascular alterations in the brain, e.g., through EV-associated miRNAs reprogramming endothelial cells as shown in experimental ALS [162,163].

There are upcoming potential and already available therapeutic strategies to tackle microvascular dysfunction in ALS. The opportunity to prevent or treat motor cortical small vessel malfunction via application of exerkines, EVs, cell-based therapies and pericyte- or senescence-specific therapeutics provides hope that new approaches will promote microvascular health and preserve motor neuron function.

## 6. Concluding Remarks on Vascular Rethinking in ALS Management

The evidence presented here allows us to conclude that in ALS, motor cortical vascular supply patterns through ACA and MCA branches could play a pivotal role for disease onset and pathology spread and thus represent new factors of disease resistance and resilience in this devastating neurodegenerative disorder. Of note, vascular supply can be considered both a causal and a modifying factor in a molecularly complex disease such as ALS, where mutually several pathomechanisms play a role [164]. Our hypothesis suggests that microvascular health mediates the effect of vascular supply on cortical motor neuron function. This is certainly attractive as it proposes upstream mechanisms supposed to precede neuronal dysfunction and degeneration, holding the potential for disease prevention and prediction. This idea is supported by emerging evidence that ALS has a significant (micro)vascular component contributing to its motor cortical degenerative presentation. Vascular supply and microvascular health could thus be considered as relevant biological processes involved in risk constellations for disease onset and progression, potentially explaining aspects of the inter-individual variance. Targeting brain (particularly motor cortical) microvascular health preventively and therapeutically might represent a powerful approach to mitigate ALS-related neurodegeneration. The link between vascular health, cerebromicrovascular integrity and synaptic or neuronal dysfunction in similar region-specific spreading neurodegenerative diseases, such as AD or PD, further highlights the importance of “(micro)vascular integration in neurodegeneration”, to go beyond a pure “neuron-centric” concept [74,165]. A spotlight on the relationship between microvascular and neuronal health is needed in ALS.

The consequent, already ongoing, identification of effective targets in the cerebral microvasculature and the NVU will become a realistic prospect for individualized therapy along with lifestyle modifications and causal cell-based treatments. The parallel refinement of novel in vivo imaging markers for depicting and monitoring brain (micro)vascular health in ALS has the potential to facilitate personalized risk stratification, e.g., indirectly through cortical supply patterns that can be determined in each individual patient. This is certainly important, as current prediction models mainly focus on clinical symptoms, which reflect already existing neuronal damage, but not necessarily presymptomatic pathology [9]. We here discuss MRI methods and sequences to depict and quantify motor cortical supply and microvascular health that can directly be related to the structural and functional characteristics, i.e., clinical, motor cortical body part-wise involvement and thus to disease/symptom onset and spread of pathology. If conducted in a longitudinal manner, MRI is supposed to detect the motor cortical stage- and body part-dependency of vascular and neuronal involvement in ALS, holding the potential for a patient-precise assessment in terms of individual progression dynamics and prognosis. This is of critical importance as there is still a lack of robust (bio)markers to foresee the so far unpredictable personalized course of disease and symptom spread, a prerequisite to pre-select and translate targeted therapies through the implementation of advanced, adaptive and multi-stage trial designs into the clinic [29,164].

Indeed, one key question is whether vascular supply patterns represent a stable individual profile that indicates local susceptibility or whether these are modifiable and dynamic, varying throughout the life and disease span. In case of the latter, one has to consider reverse causality, in which the reduced oxygen and energy requirements of motor neuron degeneration might lead to a consecutive remodeling of the supplying vasculature, which results in a decrease in diameter and thus lower MRI visibility of supplying arteries. A project within the scope of the German research foundation-funded collaborative research center 1436 (SFB 1436) is currently investigating to what extend the MRI-detected visible supply patterns are modifiable during the lifespan, comparing young vs. old and applying spatial navigation training. To address these questions, longitudinal ultra-high-resolution MRI studies accompanying ALS patients through the disease course are required.

Vascular integration has implications for the dedicated consultation and management of ALS patients, their relatives and cases with familial ALS at increased risk, e.g., due to monogenetic mutations with high penetrance (e.g., C9orf72, SOD1, TARDBP or FUS [29]). Most often, management is mainly centered around multidisciplinary symptomatic (palliative) care approaches and maximization of quality of life [29]. Maintaining vascular health is, however, not the major focus for most clinicians treating ALS patients. Our considerations highlight that vascular risk should not be ignored in ALS, not only to prevent cardiovascular disease, but to potentially impact the onset and progression of this devastating motor neuron disorder. Hence, ALS patients and those at risk should have close monitoring of vascular risk factors, in particular arterial hypertension, the main risk factor for microvascular brain damage and one of the most common comorbidities in ALS [21], and undergo appropriate medical treatment and lifestyle interventions according to current guidelines [166,167]. This includes physical activity, i.e., at least 150 min per week of mild to moderate exercise, the Mediterranean diet, avoidance of smoking, and tailored lipid-lowering therapies. As pointed out above, one recent meta-analysis showed beneficial effects of aerobic exercise in ALS, which is exercise on a treadmill, cycle ergometer, recumbent cycle or with arm-leg ergometry in two–three training sessions/week [148]. One recent meta-analysis demonstrated ALS risk reduction through several anti-hypertensives with quite similar effects among angiotensin-converting enzyme inhibitors, beta-blockers, calcium-channel blockers and diuretics, if taken regularly [37]. Calcium-channel blockers in particular might have the potential to improve cerebromicrovascular hemodynamics [69]. This is in line with the impressive effects of intense blood pressure control (systolic blood-pressure target of less than 120 mmHg), not only for cardiovascular event prevention [168] but also for preventing neurodegeneration itself, as indicated by reduced cognitive impairment through anti-hypertensive treatment [169].

Additionally, chronic psychosocial stress (CPS) and highly covariant emotional disorders such as depression are increasingly recognized modifiers of (brain) vascular health. For example, CPS and depression have not only been related to higher cardiovascular risk but also to alterations of the brain’s endothelial transcriptome and to reduction of brain microvascular perfusion [170,171]. Similarly, depression is associated with a greater risk for cognitive decline (i.e., neurodegeneration) and for higher cardiac and all-cause mortality. Therefore, depression has recently been listed as a modifiable risk factor for dementia and cardiovascular disease [19,172].

Depression, along with fatigue, dysfunctional sleep, apathy and irritability are very common in ALS and several patients likely experience CPS due to the critical life event of being confronted with a fatal diagnosis [173,174,175]. Increased depression prevalence is further highly correlated with chronic pain, a common comorbidity in ALS patients [176]. Pharmacological and non-pharmacological treatment of these symptoms should include systematic strategies to provide social support (e.g., confiding relationships, practical help and maintenance of communications skills, especially for bulbar-onset patients) and access to psychological therapy facilitating acceptance and commitment to the diagnosis [175,177]. Of note, different forms of exercise are also useful for pain prevention and treatment and through this indirectly mitigate CPS and depressive symptoms, likely with favorable effects on the cerebral microvasculature [176].

ALS patients should be informed and systematically educated about the importance of (brain) vascular health, the associated impact of vascular and mental health, and the fact that new experimental concepts are in progress that aim to maintain and restore the brain microvasculature, as pointed out above. Patients, caregivers and clinicians should closely follow developments in the (experimental) field of microvascular treatment concepts in ALS to prepare for their translational clinical application, which is just becoming a reality.

## Figures and Tables

**Figure 1 cells-12-00957-f001:**
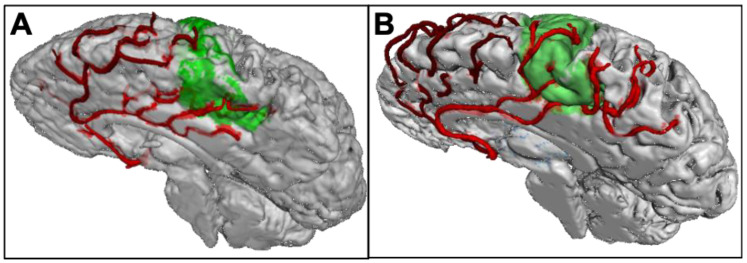
Vascular supply patterns of the motor cortex through the anterior cerebral artery in healthy young controls. (**A**) “Double supply” pattern in the medial motor cortex by branches of the pericallosal artery (light red) and callosomarginal artery (dark red). (**B**) displays a “single supply” pattern in the medial motor cortex by branches of the pericallosal artery only (light red). The motor cortex is colored green. Volunteers received 7T MPRAGE scans; vessels were manually delineated; the motor cortex was segmented using Freesurfer.

**Figure 2 cells-12-00957-f002:**
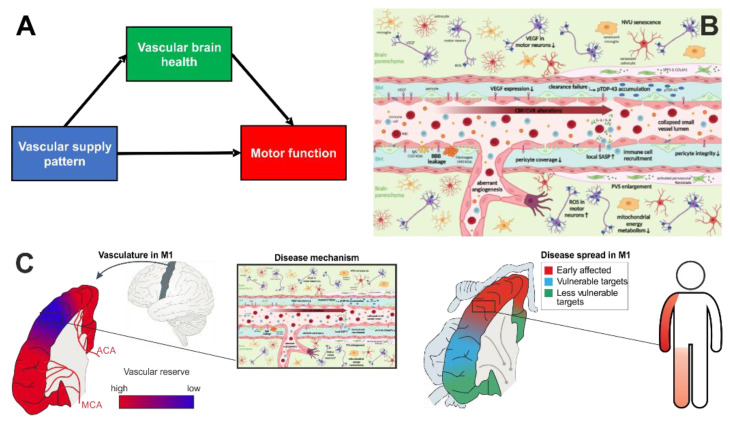
Motor cortical supply affects motor function and disease spread through microvascular brain health. (**A**) Overview of the proposed mediation model in ALS, with motor cortical supply as the independent, microvascular brain health as the mediation and motor function as the outcome variable. In (**C**), each of the variables presented in (**A**) are outlined in detail. (**C** left) Areas of lower vascular supply (in blue), putatively marked by larger distances to feeding arteries, are characterized by lower local vascular resistance and resilience, which is mediated by a cascade of microvascular alterations (**C** center, enlarged in **B**). (**C** right) Degree of motor cortical resistance and resilience determines the time course of the affection of different motor cortical areas. For example, low supply of the ipsilateral motor cortical hand area (e.g., caused by MCA single supply), promotes symptom onset in the contralateral upper limb (in red). Motor cortical regions at risk, i.e., characterized by insufficient perfusion, e.g., due to low overlap of MCA and ACA perfusion territories and subsequently affected during cortical disease spread, determine clinical disease spread to the second body part (e.g., contralateral lower limb, in light red). (**B**,**C** center) demonstrate motor cortical microvascular alterations: (i) reduced VEGF expression, (ii) BBB leaks, (iii) reduced small vessel pericyte coverage accompanied by CBF/CVR decrease and aberrant angiogenesis, (iv) activated fibroblasts found in enlarged PVS secreting SPP1 and COL6A1. Increased turnover of resident immune cells leads to decline of brain energy metabolism, NVU senescence with SASP, immune cell recruitment, finally advancing neuroinflammation and neurodegeneration. ACA, anterior cerebral artery; ALS, amyotrophic lateral sclerosis; BBB, blood–brain barrier; BM, basement membrane; CBF, cerebral blood flow; CNS, central nervous system; COL6A1, collagen VI alpha1; CVR, cerebrovascular reserve; EC, endothelial cell; IgG, immunoglobulin G; IL, interleukin; MCA, middle cerebral artery; MRI, magnetic resonance imaging; NVU, neurovascular unit; pTDP43, phosphorylated aggregates of 43 kDa transactive response DNA-binding protein; PVS, perivascular space; RBC, red blood cell; ROS, reactive oxygen species; SASP, senescence-associated secretory phenotype; SPP1, secreted phosphoprotein 1; VEGF, vascular endothelial growth factor.

**Figure 3 cells-12-00957-f003:**
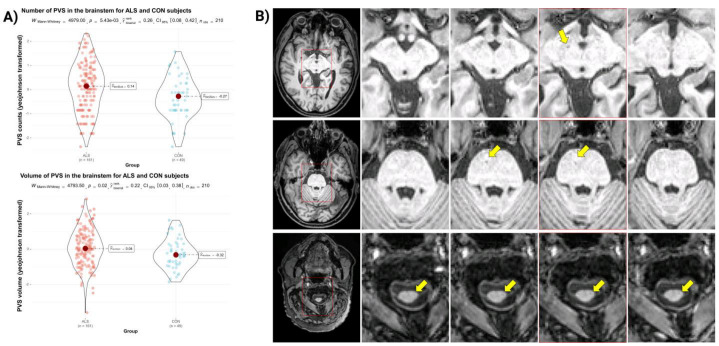
The total burden (count and volume) of PVS in the brainstem is significantly higher in subjects with ALS than in controls. We quantified PVS in a case-control study (*n* = 210; 76 female; median age 71.67 [IQR 62.80, 79.77] years; ALS/CON 161/49; T1w and FLAIR). In summary, the considered pipeline uses FreeSurfer [115] and FAST [116] for segmenting healthy and pathological white matter regions (T1w for whole brain parcellation and FLAIR for white matter hyperintensity segmentation), the Frangi filter for enhancing tubular structures (e.g., PVS), hard thresholding for segmenting PVS and connected component analysis for determining counts and volumes [117]. We transformed measurements using the Yeo–Johnson transform to deal with skewness. (**A**) In univariate analysis, we found that the count and volume of PVS in the brainstem were significantly higher in subjects with ALS than in CON (counts: Wilcoxon–Mann–Whitney = 4979, *p* = 5.43 × 10^−3^; volumes: Wilcoxon–Mann–Whitney = 4793, *p* = 0.02). The pattern remained even after controlling for age and sex (counts: 0.07 [95%-CI −0.07, 0.21] for ALS vs. −0.37 [95%-CI −0.62, −0.12] for CON, *p* = 0.003; volumes: 0.05 [95%-CI −0.09, 0.19] for ALS vs. −0.28 [95%-CI −0.54, −0.03] for CON, *p* = 0.025). (**B**) In subjects with ALS, PVS (yellow arrows) may appear enlarged in the brainstem. The original T1w image and the area we zoomed in on (red square) are both displayed in the first column. The fourth column shows the reference slice. The second and third columns are slices inferior to the reference whereas the fifth one is a slice superior to it. ALS, amyotrophic lateral sclerosis; CON, controls; FLAIR, Fluid-Attenuated Inversion Recovery; PVS, perivascular space.

## Data Availability

Data can be made available through the corresponding author upon reasonable request.

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
