# Peer review of "Brain Vascular Health in ALS Is Mediated through Motor Cortex Microvascular Integrity"

_cells, 2023, doi:10.3390/cells12060957_

Round 1

Reviewer 1 Report

The manuscript reviews the knowledge about the role of vascular factors in the development of ALS. The topic is of great importance and very actual. While the manuscript collects large amounts of data, they are not always presented clearly.

Some reasoning should be reconsidered. In the abstract it is stated that “Impaired vascular brain health has detrimental effects on motor neurons as lowered vascular endothelial growth factor levels compromise endothelial cell formation, promoting blood-brain barrier leakage and inflammation.”

The formulation and reasoning is not clear because VEGF indeed can induce endothelial cell proliferation but also induces a leaky barrier. This reasoning problem appears in the explanation of Fig. 2. as well.

„The relationship between vascular health and ALS is supported by several clinical observations and systematic studies. First, ALS patients suffer from cerebro- and cardiovascular diseases with comparable or even significantly higher frequency than control subjects.”

A comparable frequency does not support the affirmation.

The paragraph „Vascular supply of the medial temporal lobe” is a bit out of focus

It should be discussed and if possible it should be supported by literature data how is double vessel supply related to blood flow and how this affects tissue perfusion in case of non-focal vascular lesions.

Correct citation of papers is needed. E.g. it is stated that „Experimental prevention of pericyte loss slowed motor neuron degeneration …(ref 73)” In fact in the cited paper intraperitoneal pericyte injections were performed and no investigation on brain pericytes was performed.

The legend to Fig 2 is too long. Described mechanisms should be included in the text and should be supported by citations. The text in Fig 2B (middle) is not readable (at least in the version available for the reviewer).

Are there any data available about VEGF distribution in different areas of the motor cortex?

Author Response

The manuscript reviews the knowledge about the role of vascular factors in the development of ALS. The topic is of great importance and very actual.
The authors thank the referee for her/his positive evaluation of the manuscript.
While the manuscript collects large amounts of data, they are not always presented
clearly.
Some reasoning should be reconsidered. In the abstract it is stated that “Impaired vascular brain health has detrimental effects on motor neurons as lowered vascular endothelial growth factor
levels compromise endothelial cell formation, promoting blood-brain barrier leakage and inflammation.”
The formulation and reasoning is not clear because VEGF indeed can induce endothelial cell proliferation but also induces a leaky barrier. This reasoning problem appears in the explanation of Fig. 2. as well.
VEGF levels are lowered in ALS. We thus now stated in the abstract: „... vascular endothelial growth factor levels
are lowered in ALS, which can compromise endothelial cell formation and the integrity of the blood-brain barrier“. Figure legend of Figure 2 was as well changed accordingly.
„The relationship between vascular health and ALS is supported by several clinical observations and systematic studies. First, ALS patients suffer from cerebro- and cardiovascular diseases with comparable or even significantly higher frequency than control subjects.” A comparable frequency does not support the affirmation.
We apologize for this wrong statement and corrected it with reference to Diekmann et al.
(https://doi.org/10.1007/s00415-020-09799-z) as follows: „First, ALS patients suffer from cerebrovascular dis-
ease with significantly higher frequency than control subjects.“

The paragraph „Vascular supply of the medial temporal lobe” is a bit out of focus. It should be discussed and if possible it should be supported by literature data how is double vessel supply related to blood flow and how this affects tissue perfusion in case of non-focal vascular lesions.
We agree with the referee. However, while there is large evidence on hippocampal flow and (hypo)perfusion in aging and different disease conditions (e.g. arterial hypertension, for review please see Johnson DOI:10.1161/STROKEAHA.122.038263), to the best of our knowledge, there is currently no literature that particularly related double supply patterns of the medial temporal lobe to its blood flow and tissue perfusion. Our group is indeed working on exactly that question within the CRC 1436 „Neural Ressource of Cognition“ (funded by the German Research Foundation, Project-ID 425899996). We, however, just recently applied vessel distance mapping (VDM), which computes the distances of each voxel to its nearest vessel reflecting e.g. vessel density and distribution, to our cohort with available 7T ToF MRA. We uncovered, that greater values of VDM-metrics reflecting higher distances among vessels were associated with poorer cognitive outcomes. This points towards a pivotal role of a mixture of greater vessel density and distribution which probably underlies double supply patterns and could in the broadest sense considered a surrogate for adequate perfusion for maintenance of cognitive resilience. The corresponding manuscript is currently in revision and we would prefer not reporting the data in the available review.
Correct citation of papers is needed. E.g. it is stated that „Experimental prevention of pericyte loss slowed motor
neuron degeneration ...(ref 73)”. In fact in the cited paper intraperitoneal pericyte injections were performed and no investigation on brain pericytes was performed.
The authors apologize for the incorrect restitution of the cited paper content. We accordingly adapted the
sentence as follows: „Hence, in experimental ALS intraperitoneal injection of pericytes increased survival and co-culture between pericytes and motor neurons/other neuronal cells elevated the expression of genes encoding antioxidant enzymes, which both could point towards slowing of motor neuron degeneration [73].“
The legend to Fig 2 is too long. Described mechanisms should be included in the text and should be supported
by citations. The text in Fig 2B (middle) is not readable (at least in the version available for the reviewer).
We shortened the legend of Figure 2 accordingly and re-arranged the Figure, which now includes a new and
enlarged Subfigure B.
Are there any data available about VEGF distribution in different areas of the motor cortex?
To the best of our knowledge, available literature, e.g. on autopsy studies in controls and ALS or in rodents, reporting (protein) VEGF(R) levels in the (motor) cortex did not consider VEGF distribution in different areas of the motor cortex (e.g. Brockington et al. doi: 10.1097/01.jnen.0000196134.51217.74 or Hou et al.
doi:10.1016/j.jchemneu.2011.06.001; for review e.g. Lange et al. doi:10.1038/nrneurol.2016.88).

Reviewer 2 Report

This is a very interesting paper dealing with emerging evidence of the role of neurovascular unit and brain vascular health in ALS pathogenesis, progression and as possible therapeutic target. 

The paper is very clear, well organized, well written and examines comprehensively several research issues on the topic.  I would suggest only to smoothen the sections related to potential treatments, especially those related to cell-based therapy, given that till now they failed to achieve any effectiveness on ALS progression/survival. 

Author Response

This is a very interesting paper dealing with emerging evidence of the role of neurovascular unit and brain vascular health in ALS pathogenesis, progression and as possible therapeutic target. The paper is very clear, well organized, well written and examines comprehensively several research issues on the topic.
We thank the referee for the very positive feedback.
I would suggest only to smoothen the sections related to potential treatments, especially those related to cell-based therapy, given that till now they failed to achieve any effectiveness on ALS progression/survival.
We accordingly smoothed the section on „Targeting brain vascular health“, certainly the part related to cell- based therapy.